# The Organization of Home Palliative Cancer Care by Primary Health Care: A Systematic Review Protocol

**DOI:** 10.3390/ijerph20065085

**Published:** 2023-03-14

**Authors:** Marcelle Miranda da Silva, Thayna Barros, Cristina Lavareda Baixinho, Andreia Costa, Eunice Sá, Maria Adriana Henriques

**Affiliations:** 1Department of Nursing Methodology, Escola de Enfermagem Anna Nery, Universidade Federal do Rio de Janeiro, Rio de Janeiro 20221-110, RJ, Brazil; 2Nursing Research, Innovation and Development Centre of Lisbon (CIDNUR), 1600-190 Lisbon, Portugal; 3Nursing School of Lisbon, 1600-190 Lisbon, Portugal; 4Instituto de Saúde Ambiental (ISAMB), Faculty of Medicine, Universidade de Lisboa, 1649-028 Lisbon, Portugal; 5Laboratório Associado TERRA, Faculty of Medicine, Universidade de Lisboa, 1649-028 Lisbon, Portugal

**Keywords:** palliative care, neoplasms, home care services, primary health care, organization and administration, patient care management

## Abstract

Greater longevity and chronic diseases, such as cancer, require (re)organization of care for the sustainability of health systems and better quality of life. Palliative care organized by primary health care has positive outcomes, changing standards of care at the end of life, reducing hospitalizations and health costs and contributing to people’s autonomy to stay at home with controlled symptoms. However, in many countries, this is not possible because the provision of palliative cancer care is isolated or generalized, concentrated in the hospital, and without the strategic participation of primary care. In many developed countries, where palliative care is offered in an integrated way, home care has increased the people’s chances of receiving dignified care at the end of life. The aim of this review is to evaluate the organization of home palliative cancer care by primary care to improve the use of health resources and the quality of life of such patients. This systematic review protocol follows the Cochrane methodology to provide a narrative synthesis, with the resulting report guided by the Preferred Reporting Items for Systematic Reviews (PRISMA).

## 1. Introduction

Palliative care “is the active holistic care of individuals across all ages with serious health-related suffering due to severe illness, and especially of those near the end of life” [1]. Palliative care deals with “prevention, early identification, comprehensive assessment and management of physical issues, including pain and other distressing symptoms, psychological distress, spiritual distress and social needs”. Thus, the assistance of the interdisciplinary team seeks to promote comfort and quality of life and support family caregivers, wherever they are [1].

To achieve these aims, the World Health Organization guides the early integration of palliative care from the time of diagnosis of a disease with the potential to threaten the continuity of life. In the integration of clinical and palliative care, the evolution of investments in palliative care occurs in response to people’s needs and expectations. However, when the disease does not respond to therapies for cure or control, palliative care gains greater relevance [1,2].

However, palliative care is often insufficient and precarious in low- and middle-income countries, due, for example, to deficits in the education of health professionals and the general public on this topic, a lack of resources, and the inability of public–private health systems to provide palliative care with quality and safety [3,4].

Many people do not receive palliative care, and the experience of death involves a lot of suffering and neglect [2,5]. The difficulties in performing palliative care are rooted, for example, in the biomedical model, which implies the non-acceptance of therapeutic failures for cure, late recognition of palliative care patients, and unrealistic prognoses [5]. Faced with these limitations, palliative care is more acceptable among people with cancer compared to other people with diseases with a longer course and difficult prognosis [6]. In addition to its epidemiological importance, this is the main reason limiting the present review to the field of oncology.

In advanced cancer, clinical instability and the exacerbation of symptoms can lead to the use of urgencies and emergencies, as well as long hospital stays, especially when care is not integrated with a specialized palliative care team for referral of complex cases. In general hospitals, suffering often continues in the face of isolation, with poor management of symptoms and possibly futile treatments, in addition to death being quite frequent in this scenario. Models centered on hospitals are associated with high financial costs due to a lack of in-hospital teams, as well as transitional and home palliative care, that is, a structured network for managing cases of people with palliative care needs [7,8].

Many developed countries deal with the rising costs of caring for people with chronic illnesses by investing in structured palliative care services, which are usually organized by primary care, understanding that palliative care is inseparable from primary care [2]. Both guided by the same principles, such as “equity, solidarity, social justice, universal access to services, multisectoral action, decentralization and community participation” ([2] p. 22). This palliative care approach in primary care includes changes in end-of-life care patterns, shifting the environment from hospital to home care [9,10].

Home care is the main modality of palliative care, as it seeks to guarantee the principles of bioethics, such as patient autonomy [8]. It is the way to reach more people who need palliative care [2]. For home palliative care to be viable, the presence of a family caregiver is necessary, who will provide care at home, with full professional guidance [9]. The health team performs home consultations according to demand, and nurses commonly coordinate this care as part of an interdisciplinary team [9]. 

The organization of home palliative cancer care by the primary care team can follow different frameworks to provide generalist palliative care and specialized palliative care and meet the needs of patients and families [2,9,10]. Care models can vary depending on the health system, providing specialized palliative care at home, in a hospice program, and/or in a hospital. However, the participation of primary care professionals is essential to improve integration with specialized palliative care in any model [10]. 

In-home palliative cancer care is mandatory to prescribe and manage drugs, as well as non-pharmacological interventions to control physical and psychological symptoms; identify patients at the end of life; provide support in the case of death at home and during bereavement; provide nursing care for comfort; offer assistance through phone calls and be prepared for anticipatory visits in order to reduce emergences admissions and hospitalizations; and review all necessary resources to qualify the care [2,8].

This literature review shows that no evidence synthesis studies have been conducted with a focus on the organization of home palliative cancer care by primary care. The knowledge is heterogeneous and fragmented, and a systematic review can contribute to guiding many countries that urgently need to increase their capacity to offer palliative care. In addition to responses relating to costs, the evidence needs to be strengthened with regard to the potential to improve the quality of life of palliative care patients [11].

Therefore, we aimed to evaluate the organization of home palliative cancer care by primary care to improve the use of health resources and the quality of life of such patients. 

## 2. Methods

A systematic review of the literature will be performed based on the following research question: How does home palliative cancer care organized by primary care impact the management of health resources and people’s quality of life? The protocol of this study followed the Cochrane Handbook for Systematic Reviews of Interventions [12].

This systematic review was registered in the PROSPERO database (registration number: CRD42023392997) [13], and the resulting report will be guided by the Preferred Reporting Items for Systematic Reviews (PRISMA) [14].

### 2.1. Information Sources and Search Strategy

The academic databases used to retrieve the studies will be MEDLINE (via PubMed), Scopus (through the Elsevier interface), LILACS, Web of Science, Cochrane Library, and CINAHL (via EBSCOhost). Studies published from 2018 to 2022 will be included; this timeframe is justified by the need to capture contemporary practice according the 2018 definition of palliative care of the International Association for Hospice and Palliative Care [1].

Bibliographies of primary studies and review articles meeting the inclusion criteria will be searched manually to identify further eligible studies.

The search strategy will include terms related to palliative cancer care at home and home care organized by primary care, and we will include all interventions that enable care at home to improve the quality of life of such patients, as well as resource use, considering the public–private health system. 

The search strategy will be adapted for each database in combination with its specific filters. Table 1, Table 2, Table 3, Table 4, Table 5 and Table 6 present this strategy for each database. There will be no language limit. The search will be rerun prior to the final analysis. We will not include any unpublished data.

### 2.2. Eligibility Criteria

Studies will be included based on participants, interventions, context, outcomes, and type of study. 

#### 2.2.1. Participants

Studies will include adult participants. There will be no patient limits in terms of gender, ethnicity, type of cancer, and the length of in-home care. There will also be no limits for health professionals involved in terms of professional category, function, and knowledge of palliative care. There will be no limits on other participants such as family caregivers, volunteers, religious representatives, or other members of the general population. We will accept studies in which interventions were carried out exclusively at the home of the patient, whether in primary health care or otherwise. Pediatric studies and those that address chronic diseases other than cancer or that do not report data for cancer care separately will be excluded.

#### 2.2.2. Interventions

We will include studies evaluating:-Interventions for organizing home care services;-Interventions to integrate primary care into the care network for patients in palliative cancer care;-Interventions describing palliative care programs at home with the aim of improving quality of life, promoting comfort, controlling symptoms, relieving suffering, preserving the dignity of the person, and preparing for grief;-Interventions to improve the management of health system resources, focusing on the frequency of home consultations, the time taken to manage cases in palliative care at home, use of urgencies and emergencies, hospitalizations, and death at home;-Interventions taking into account quality criteria in palliative care.

#### 2.2.3. Context

We will include studies carried out at patients’ homes and monitored by healthcare teams as a result of advanced cancer in palliative care. Countries will be included regardless of the level of economic development, from those with no knowledge of palliative care activities to those with advanced and integrated palliative care, according to the classification of the Global Atlas of Palliative Care [5].

#### 2.2.4. Outcomes

All outcome measures will be extracted according to interventions of interest and inclusion criteria for this review.

Primary outcome measures will include management of healthcare resources, relevance of primary care in palliative care, quality of life, relief from suffering, symptom control, duration of the stay at home, use of urgency and emergency services, death at home, and the grieving process.

Secondary outcome measures will include palliative care education, the level of integration between generalist palliative care and specialized palliative care, community support networks, economic data, and policy issues for the implementation of palliative care.

#### 2.2.5. Types of Studies

We will include experimental (e.g., randomized control trials and crossover trials) and observational studies (e.g., cohort and case–control studies). 

### 2.3. Data Extraction (Selection and Coding)

EndNote^®^ reference manager [15] will be used to automatically remove duplicates; subsequently, data will be recorded in Microsoft Office EXCEL^®^ 2019 (Microsoft Corporation, Redmond, WA, USA) spreadsheets. First, all the titles and abstracts of the identified articles will be analyzed in relation to the established inclusion criteria, with the aim of assessing the eligibility of the studies. Four team members will participate in the abstract-level eligibility review (M.M.d.S., T.B., C.L.B., and E.S.), and each abstract will be independently reviewed by two team members (M.M.d.S. and T.B.). The same procedure will be implemented at the full-text level. Data will be extracted from the selected articles in online format by two blinded team members (M.M.d.S. and T.B.), and a third member will verify the extracted data (C.L.B.). Disagreements at all stages will be resolved by consensus. 

Studies considered relevant will be selected for further reading and included in the review. The reasons for exclusion of studies will also be documented.

Study investigators will be contacted by e-mail for unreported data or additional details. Data synthesis will be operationalized by quantification of themes and shared patterns.

We will extract the following variables: study characteristics (duration, location and environment, type of study, population, and sample), ways of implementing home palliative cancer care in health systems, outcomes related to resource management, quality and integration of care, and reported bias or limitations. Information on the categorization of palliative care development in the country (stratification from 1 to 4b) will also be collected according to the Global Atlas of Palliative Care [5].

Other information about data extraction will be recorded from reports (name of data extractors, date of data extraction, and identification features of each report), in addition to eligibility criteria (confirming eligibility or justifying exclusion).

#### Risk of Bias (Quality) Assessment

Two reviewers (M.M.d.S. and T.B.) will independently assess the risk of bias of the included studies. We will consider the following instruments: the Cochrane risk-of-bias tool for randomized trials (RoB 2), the Cochrane risk-of-bias tool for non-randomized studies—of interventions (ROBINS-I), and the Cochrane tool for assessing risk of bias due to missing evidence in a synthesis (ROB ME) [13].

Any disagreements between the two reviewers will be resolved by discussion, with the involvement of a third review author (C.L.B.). The level of risk of bias (low, high, and some concerns) in each of these domains will be presented separately for each study.

### 2.4. Strategy for Data Synthesis

The following data will be synthesized: time spent managing care at home, rate of use of urgent and emergency services during home care, the number of deaths at home, the number of home consultations by the professional category, applied resources (material and human), means of communication and integration of the service with the health network involving all interested parties, patient performance status and symptom control scores, quality of life assessment scores, the number of families served by a given team, and the relationship between consultations carried out by the generalist palliative care/primary healthcare team and the specialized palliative care team.

We will carry out a narrative synthesis of the results according to the following characteristics:-Type of intervention; for example, organization of services, integration into the network, and participation in primary care;-Characteristics of patients’ in-home care, family, and community profile, such as age, gender, ethnicity, socioeconomic status, educational level, low-/middle-/high-income country (classification according to the Human Development Index of the United Nations Development Program) [16];-Type of outcome, e.g., symptom control and rate of deaths at home;-Intervention content, number of consultations, health professionals involved, number of telephone calls and teleconsultations, and duration of case management in home care.

## 3. Discussion

According to global recommendations, palliative care needs to be strengthened and integrated into all health systems [1,2,5]. Palliative care is an urgent humanitarian need, given its potential to alleviate physical, psychosocial, and spiritual problems in more than 90% of people with advanced cancer [5]. The provision of palliative care must be available wherever the person is [5,8]. Many patients with advanced cancer choose to die at home, but the lack of services to support this decision contributes to the high use of urgency and emergency services, as well as death in the hospital [2,9,10,17]. The potential of primary care is promising and needs to be better used in relation to the management of cancer in the community context [18].

Thus, this systematic review will provide important information for the organization of services, expansion of care capabilities, and integration of care through greater participation of primary care in the context of cancer diseases, with the patient’s home as the main place for care.

## 4. Conclusions

Provision of quick and creative responses to the needs of people at the end of life is necessary, given the economic impact caused by cancer and other chronic diseases. This review will summarize evidence that may enable the expansion of palliative cancer care in primary care, namely in-home care, in alignment with the advances of public policies in the global context and the aims of sustainable development. To this end, in addition to the scientific rigor that will be adopted in this review in terms of dissemination, the authors plan to publish a final report in a journal indexed in international databases.

## Figures and Tables

**Table 1 ijerph-20-05085-t001:** Search strategy in MEDLINE.

Acronyms	Search Strategy	Results
#1 Participants	(((((((((((((((((((((((((((((((“Palliative Care”) OR (“Palliative Treatment”)) OR (“Palliative Treatments”)) OR (“Palliative Therapy”)) OR (“Palliative Supportive Care”)) OR (“Palliative Medicine”)) OR (“Palliative Care Medicine”)) OR (“Hospice Care”)) OR (“Hospice Programs”)) OR (“Hospice Program”)) OR (“Bereavement Care”)) OR (“Hospice and Palliative Care Nursing”)) OR (“Palliative Nursing”)) OR (“Palliative Care Nursing”)) OR (“Hospice Nursing”)) OR (“Hospice Nursings”)) AND (neoplasms)) OR (neoplasm)) OR (tumor)) OR (tumors)) OR (cancer)) OR (cancers)) OR (“Malignant Neoplasm”)) OR (malignancy)) OR (malignancies)) OR (“Malignant Neoplasms”)) OR (“Palliative Cancer Care”)) OR (adult)) OR (aged)) OR (elderly)) NOT (child)) NOT (children)	2,260,072
#2 Intervention	((((((((((((((((((((((((((((((((“Home Nursing”) OR (“Nonprofessional Home Care”)) OR (“Non-Professional Home Care”)) OR (“Home Care Services”)) OR (“Home Care Service”)) OR (“Domiciliary Care”)) OR (“Home Health Care”)) OR (“Home Care”)) OR (“Home Care Agencies”)) OR (“Home Care Agency”)) OR (“Home Health Care Agencies”)) OR (“Home Health Agencies”)) OR (“Home Health Agency”)) OR (“Nursing Services”)) OR (“Nursing Service”)) OR (“Home Health Nursing”)) OR (“Home Health Care Nursing”)) OR (“Home Health Aides”)) OR (“Home Health Aide”)) OR (“Homemaker-Home Health Aides”)) OR (“Homemaker Home Health Aides”)) OR (“Homemaker Home Health Aide”)) OR (“Home Care Aides”)) OR (“Home Care Aide”)) OR (“Primary Health Care”)) OR (“Care, Primary Health”)) OR (“Health Care, Primary”)) OR (“Primary Healthcare”)) OR (“Healthcare, Primary”)) OR (“Primary Care”)) OR (“Care, Primary”)) OR (“Public Health Practice”)) OR (“Public Health Practices”)	102,437
#3 Outcomes	((((((((((((((((((((((((((((((((((((“Quality of Life”) OR (“Life Quality”)) OR (“Health-Related Quality Of Life”)) OR (“Health Related Quality Of Life”)) OR (hrqol)) OR (“Patient Acceptance of Health Care”)) OR (“Health Care Utilization”)) OR (“Patient Acceptance of Healthcare”)) OR (“Healthcare Patient Acceptance”)) OR (“Healthcare Patient Acceptances”)) OR (“Nonacceptors of Health Care”)) OR (“Health Care Nonacceptor”)) OR (“Health Care Nonacceptors”)) OR (“Health Care Seeking Behavior”)) OR (“Acceptors of Health Care”)) OR (“Health Care Acceptor”)) OR (“Health Care Acceptors”)) OR (“Acceptability of Health Care”)) OR (“Health Care Acceptability”)) OR (“Acceptability of Healthcare”)) OR (“Healthcare Acceptabilities”)) OR (“Healthcare Acceptability”)) OR (“Organization and Administration”)) OR (“Administration and Organization”)) OR (logistics)) OR (supervision)) OR (administration)) OR (“Administrative Coordination”)) OR (“Disease Management”)) OR (“Pain Management”)) OR (“Patient Care Management”)) OR (“Financial Management”)) OR (funds)) OR (endowments)) OR (“Financial Activity”)) OR (“Population Health Management”)) OR (“Population Health Managements”)	1,413,929
#4 Study design to be included	(((((((((((((((((((((“Non-Randomized Controlled Trials as Topic”) OR (“Non Randomized Controlled Trials as Topic”)) OR (“Quasi-Experimental Studies”)) OR (“Quasi Experimental Studies”)) OR (“Quasi-Experimental Study”)) OR (“Nonrandomized Clinical Trial”)) OR (“Nonrandomized Clinical Trials”)) OR (“Non-Randomized Clinical Trial”)) OR (“Non-Randomized Clinical Trials”)) OR (“Non-Randomized Clinical Trials”)) OR (“Nonrandomized Controlled Trials as Topic”)) OR (“Randomized Controlled Trials as Topic”)) OR (“Clinical Trials, Randomized”)) OR (“Trials, Randomized Clinical”)) OR (“Controlled Clinical Trials, Randomized”)) OR (“Observational Studies as Topic”)) OR (“Observational Study as Topic”)) OR (“Evaluations of Interventions”)) OR (“Observational Studies”)) OR (“Observational Study”)) OR (“Randomized Controlled Trials”)) OR (“Non-Randomized Controlled Trials”)	1,028,089
#1 AND #2 AND #3 AND #4		7929

**Table 2 ijerph-20-05085-t002:** Search strategy in Scopus.

Acronyms	Search Strategy	Results
#1 Participants	(TITLE-ABS-KEY (“Palliative Care”) OR TITLE-ABS-KEY (“Palliative Treatment”) OR TITLE-ABS-KEY (“Palliative Treatments”) OR TITLE-ABS-KEY (“Palliative Therapy”) OR TITLE-ABS-KEY (“Palliative Supportive Care”) OR TITLE-ABS-KEY (“Palliative Medicine”) OR TITLE-ABS-KEY (“Palliative Care Medicine”) OR TITLE-ABS-KEY (“Hospice Care”) OR TITLE-ABS-KEY (“Hospice Programs”) OR TITLE-ABS-KEY (“Hospice Program”) OR TITLE-ABS-KEY (“Bereavement Care”) OR TITLE-ABS-KEY (“Hospice and Palliative Care Nursing”) OR TITLE-ABS-KEY (“Palliative Nursing”) OR TITLE-ABS-KEY (“Palliative Care Nursing”) OR TITLE-ABS-KEY (“Hospice Nursing”) OR TITLE-ABS-KEY (“Hospice Nursings”) AND TITLE-ABS-KEY (neoplasms) OR TITLE-ABS-KEY (neoplasm) OR TITLE-ABS-KEY (tumor) OR TITLE-ABS-KEY (tumors) OR TITLE-ABS-KEY (cancer) OR TITLE-ABS-KEY (cancers) OR TITLE-ABS-KEY (“Malignant Neoplasm”) OR TITLE-ABS-KEY (malignancy) OR TITLE-ABS-KEY (malignancies) OR TITLE-ABS-KEY (“Malignant Neoplasms”) OR TITLE-ABS-KEY (“Palliative Cancer Care”) OR TITLE-ABS-KEY (adult) OR TITLE-ABS-KEY (aged) OR TITLE-ABS-KEY (elderly) AND NOT TITLE-ABS-KEY (child) AND NOT TITLE-ABS-KEY (children)) AND PUBYEAR > 2017 AND PUBYEAR > 2017	25,256
#2 Intervention	(TITLE-ABS-KEY (“Home Nursing”) OR TITLE-ABS-KEY (“Nonprofessional Home Care”) OR TITLE-ABS-KEY (“Non-Professional Home Care”) OR TITLE-ABS-KEY (“Home Care Services”) OR TITLE-ABS-KEY (“Home Care Service”) OR TITLE-ABS-KEY (“Domiciliary Care”) OR TITLE-ABS-KEY (“Home Health Care”) OR TITLE-ABS-KEY (“Home Care”) OR TITLE-ABS-KEY (“Home Care Agencies”) OR TITLE-ABS-KEY (“Home Care Agency”) OR TITLE-ABS-KEY (“Home Health Care Agencies”) OR TITLE-ABS-KEY (“Home Health Agencies”) OR TITLE-ABS-KEY (“Home Health Agency”) OR TITLE-ABS-KEY (“Nursing Services”) OR TITLE-ABS-KEY (“Nursing Service”) OR TITLE-ABS-KEY (“Home Health Nursing”) OR TITLE-ABS-KEY (“Home Health Care Nursing”) OR TITLE-ABS-KEY (“Home Health Aides”) OR TITLE-ABS-KEY (“Home Health Aide”) OR TITLE-ABS-KEY (“Homemaker-Home Health Aides”) OR TITLE-ABS-KEY (“Homemaker Home Health Aides”) OR TITLE-ABS-KEY (“Homemaker Home Health Aide”) OR TITLE-ABS-KEY (“Home Care Aides”) OR TITLE-ABS-KEY (“Home Care Aide”) OR TITLE-ABS-KEY (“Primary Health Care”) OR TITLE-ABS-KEY (“Care, Primary Health”) OR TITLE-ABS-KEY (“Health Care, Primary”) OR TITLE-ABS-KEY (“Primary Healthcare”) OR TITLE-ABS-KEY (“Healthcare, Primary”) OR TITLE-ABS-KEY (“Primary Care”) OR TITLE-ABS-KEY (“Care, Primary”) OR TITLE-ABS-KEY (“Public Health Practice”) OR TITLE-ABS-KEY (“Public Health Practices”))	350,316
#3 Outcomes	(TITLE-ABS-KEY (“Quality of Life”) OR TITLE-ABS-KEY (“Life Quality”) OR TITLE-ABS-KEY (“Health-Related Quality Of Life”) OR TITLE-ABS-KEY (“Health Related Quality Of Life”) OR TITLE-ABS-KEY (hrqol) OR TITLE-ABS-KEY (“Patient Acceptance of Health Care”) OR TITLE-ABS-KEY (“Health Care Utilization”) OR TITLE-ABS-KEY (“Patient Acceptance of Healthcare”) OR TITLE-ABS-KEY (“Healthcare Patient Acceptance”) OR TITLE-ABS-KEY (“Healthcare Patient Acceptances”) OR TITLE-ABS-KEY (“Nonacceptors of Health Care”) OR TITLE-ABS-KEY (“Health Care Nonacceptor”) OR TITLE-ABS-KEY (“Health Care Nonacceptors”) OR TITLE-ABS-KEY (“Health Care Seeking Behavior”) OR TITLE-ABS-KEY (“Acceptors of Health Care”) OR TITLE-ABS-KEY (“Health Care Acceptor”) OR TITLE-ABS-KEY (“Health Care Acceptors”) OR TITLE-ABS-KEY (“Acceptability of Health Care”) OR TITLE-ABS-KEY (“Health Care Acceptability”) OR TITLE-ABS-KEY (“Acceptability of Healthcare”) OR TITLE-ABS-KEY (“Healthcare Acceptabilities”) OR TITLE-ABS-KEY (“Healthcare Acceptability”) OR TITLE-ABS-KEY (“Organization and Administration”) OR TITLE-ABS-KEY (“Administration and Organization”) OR TITLE-ABS-KEY (logistics) OR TITLE-ABS-KEY (supervision) OR TITLE-ABS-KEY (administration) OR TITLE-ABS-KEY (“Administrative Coordination”) OR TITLE-ABS-KEY (“Disease Management”) OR TITLE-ABS-KEY (“Pain Management”) OR TITLE-ABS-KEY (“Patient Care Management”) OR TITLE-ABS-KEY (“Financial Management”) OR TITLE-ABS-KEY (funds) OR TITLE-ABS-KEY (endowments) OR TITLE-ABS-KEY (“Financial Activity”) OR TITLE-ABS-KEY (“Population Health Management”) OR TITLE-ABS-KEY (“Population Health Managements”))	4,353,144
#4 Study design to be included	(TITLE-ABS-KEY (“Non-Randomized Controlled Trials as Topic”) OR TITLE-ABS-KEY (“Non Randomized Controlled Trials as Topic”) OR TITLE-ABS-KEY (“Quasi-Experimental Studies”) OR TITLE-ABS-KEY (“Quasi Experimental Studies”) OR TITLE-ABS-KEY (“Quasi-Experimental Study”) OR TITLE-ABS-KEY (“Nonrandomized Clinical Trial”) OR TITLE-ABS-KEY (“Nonrandomized Clinical Trials”) OR TITLE-ABS-KEY (“Non-Randomized Clinical Trial”) OR TITLE-ABS-KEY (“Non-Randomized Clinical Trials”) OR TITLE-ABS-KEY (“Non-Randomized Clinical Trials”) OR TITLE-ABS-KEY (“Nonrandomized Controlled Trials as Topic”) OR TITLE-ABS-KEY (“Randomized Controlled Trials as Topic”) OR TITLE-ABS-KEY (“Clinical Trials, Randomized”) OR TITLE-ABS-KEY (“Trials, Randomized Clinical”) OR TITLE-ABS-KEY (“Controlled Clinical Trials, Randomized”) OR TITLE-ABS-KEY (“Observational Studies as Topic”) OR TITLE-ABS-KEY (“Observational Study as Topic”) OR TITLE-ABS-KEY (“Evaluations of Interventions “) OR TITLE-ABS-KEY (“Observational Studies”) OR TITLE-ABS-KEY (“Observational Study”) OR TITLE-ABS-KEY (“Randomized Controlled Trials”) OR TITLE-ABS-KEY (“Non-Randomized Controlled Trials”)) AND PUBYEAR > 2017 AND PUBYEAR > 2017	451,676
#1 AND #2 AND #3 AND #4		112

**Table 3 ijerph-20-05085-t003:** Search strategy in LILACS.

Acronyms	Search Strategy	Results
#1 AND #2 AND #3 AND #4	(“Palliative Care”) OR (“Palliative Treatment*”) OR (“Palliative Therapy”) OR (“Palliative Supportive Care”) OR (“Palliative Medicine”) OR (“Palliative Care Medicine”) OR (“Hospice Care”) OR (“Hospice Program*”) OR (“Bereavement Care”) OR (“Hospice and Palliative Care Nursing”) OR (“Palliative Nursing”) OR (“Palliative Care Nursing”) OR (“Hospice Nursing*”) AND (neoplasm*) OR (tumor*) OR (cancer*) OR (“Malignant Neoplasm+*) OR (malignanc*) OR (“Palliative Cancer Care”) OR (adult) OR (aged) OR (elderly) AND NOT (child*) AND (“Home Nursing”) OR (“Nonprofessional Home Care”) OR (“Non-Professional Home Care”) OR (“Home Care Service*”) OR (“Domiciliary Care”) OR (“Home Health Care”) OR (“Home Care”) OR “Home Care Agenc*”) OR (“Home Health Care Agenc*”) OR (“Nursing Service”) OR (“Home Health Nursing”) OR (“Home Health Care Nursing”) OR (“Primary Health Care”) OR (“Care, Primary Health”) OR (“Primary Care”) OR (“Public Health Practice”) AND (“Quality of Life”) OR (“Life Quality”) OR (“Health-Related Quality”) OR (hrqol) OR (“Patient Acceptance of Health Care”) OR (“Health Care Utilization”) OR (“Patient Acceptance of Health Care”) OR (“Health Care Nonacceptor”) OR (“Acceptors of Health Care”) OR Healtcare Accetabilit*”) OR (Organization and Administration) OR (logistics) OR (supervision) OR (administration) OR (“Disease Management”) OR (“Pain Management”) OR (“Patient Care Management”) OR (“Financial Management”) OR (funds) OR (“Population Health Management”)	6

**Table 4 ijerph-20-05085-t004:** Search strategy in Web of Science.

Acronyms	Search Strategy	Results
#1 Participants	((((((((((((((((((((ALL = (“Palliative Care”)) OR ALL = (“Palliative Treatment”)) OR ALL = (“Palliative Supportive Care”)) OR ALL = (“Palliative Medicine”)) OR ALL = (“Hospice Care”)) OR ALL = (“Hospice Program”)) OR ALL = (“Bereavement Care”)) OR ALL = (“Hospice and Palliative Care Nursing”)) OR ALL = (“Palliative Nursing”)) OR ALL = (“Palliative Care Nursing”)) OR ALL = (“Hospice Nursing*”)) AND ALL = (neoplasm*)) OR ALL = (tumor*)) OR ALL = (cancer*)) OR ALL = (“Malignant Neoplasm”)) OR ALL = (malignanc*)) OR ALL = (“Palliative Cancer Care”)) OR ALL = (adult)) OR ALL = (aged)) OR ALL = (elderly)) NOT ALL = (child*)	2,492,699
#2 Intervention	(((((((((((((((((((((((((ALL = (“Home Nursing”)) OR ALL = (“Nonprofessional Home Care”)) OR ALL = (“Non-Professional Home Care”)) OR ALL = (“Home Care Service*”)) OR ALL = (“Domiciliary Care”)) OR ALL = (“Home Health Care”)) OR ALL = (“Home Care”)) OR ALL = (“Home Care Agenc*”)) OR ALL = (“Home Health Care Agencies”)) OR ALL = (“Home Health Agenc*”)) OR ALL = (“Nursing Service*”)) OR ALL = (“Home Health Nursing”)) OR ALL = (“Home Health Care Nursing”)) OR ALL = (“Home Health Aide*”)) OR ALL = (“Homemaker-Home Health Aide*”)) OR ALL = (“Homemaker Home Health Aides”)) OR ALL = (“Home Care Aide*”)) OR ALL = (“Primary Health Care”)) OR ALL = (“Care, Primary Health”)) OR ALL = (“Health Care, Primary”)) OR ALL = (“Primary Healthcare”)) OR ALL = (“Healthcare, Primary”)) OR ALL = (“Primary Care”)) OR ALL = (“Care, Primary”)) OR ALL = (“Public Health Practic*”))	113,833
#3 Outcomes	(((((((((((((((((((((((((((((ALL = (“Quality of Life”)) OR ALL = (“Health Related Quality Of Life”)) OR ALL = (hrqol)) OR ALL = (“Patient Acceptance of Health Care”)) OR ALL = (“Health Care Utilization”)) OR ALL = (“Healthcare Patient Acceptance”)) OR ALL = (“Nonacceptors of Health Care”)) OR ALL = (“Health Care Nonacceptor”)) OR ALL = (“Health Care Nonacceptors”)) OR ALL = (“Health Care Seeking Behavior”)) OR ALL = (“Acceptors of Health Care”)) OR ALL = (“Health Care Acceptor*”)) OR ALL = (“Acceptability of Health Care”)) OR ALL = (“Health Care Acceptability”)) OR ALL = (“Healthcare Acceptabilit*”)) OR ALL = (“Organization and Administration”)) OR ALL = (“Administration and Organization”)) OR ALL = (logistics)) OR ALL = (supervision)) OR ALL = (administration)) OR ALL = (“Administrative Coordination”)) OR ALL = (“Disease Management”)) OR ALL = (“Pain Management”)) OR ALL = (“Patient Care Management”)) OR ALL = (“Financial Management”)) OR ALL = (funds)) OR ALL = (endowments)) OR ALL = (“Financial Activity”)) OR ALL = (“Population Health Management*”))	1,563,734
#4 Study design to be included	(((((((((((((((((((((ALL = (“Non-Randomized Controlled Trials as Topic”)) OR ALL = (“Non Randomized Controlled Trials as Topic”)) OR ALL = (“Quasi-Experimental Studies”)) OR ALL = (“Quasi Experimental Studies”)) OR ALL = (“Quasi-Experimental Study”)) OR ALL = (“Nonrandomized Clinical Trial”)) OR ALL = (“Nonrandomized Clinical Trials”)) OR ALL = (“Non-Randomized Clinical Trial”)) OR ALL = (“Non-Randomized Clinical Trials”)) OR ALL = (“Non-Randomized Clinical Trials”)) OR ALL = (“Nonrandomized Controlled Trials as Topic”)) OR ALL = (“Randomized Controlled Trials as Topic”)) OR ALL = (“Clinical Trials, Randomized”)) OR ALL = (“Trials, Randomized Clinical”)) OR ALL = (“Controlled Clinical Trials, Randomized”)) OR ALL = (“Observational Studies as Topic”)) OR ALL = (“Observational Study as Topic”)) OR ALL = (“Evaluations of Interventions”)) OR ALL = (“Observational Studies”)) OR ALL = (“Observational Study”)) OR ALL = (“Randomized Controlled Trials”)) OR ALL = (“Non-Randomized Controlled Trials”)	129,532
#1 AND #2 AND #3 AND #4		665

**Table 5 ijerph-20-05085-t005:** Search strategy in Cochrane.

Acronyms	Search Strategy	Results
#1 Participants	“Palliative Care” OR “Hospice Care” OR (“Hospice and Palliative Care Nursing”) OR “Palliative Care Nursing” OR “Palliative Medicine” AND neoplasms OR tumor OR “Palliative Cancer Care” OR adult OR aged NOT child*	335,428
#2 Intervention	“Home Nursing” OR “Home Care Services” OR “Domiciliary Care” OR “Home Care” OR “Home Care Agencies” OR “Nursing Services” OR “Primary Health Care” OR “Primary Care” OR “Home Health Aide” OR “Public Health Practice” OR “Health Care, Primary” OR “Nonprofessional Home Care”	9745
#3 Outcomes	“Quality of Life” OR “Health Related Quality Of Life” OR hrqol OR “Patient Acceptance of Health Care” OR “Health Care Acceptors” OR (“Organization and Administration”) OR “Disease Management” OR “Pain Management” OR “Financial Management” OR “Population Health Management” OR logistics OR supervision	76,213
#4 Study design to be included	“Non Randomized Controlled Trials as Topic” OR “Quasi-Experimental Study” OR “Non-Randomized Clinical Trial” OR “Randomized Controlled Trials as Topic” OR “Clinical Trials, Randomized” OR “Observational Study as Topic” OR “Observational Study” OR “Randomized Controlled Trials” OR “Non-Randomized Controlled Trials” OR “Observational Studies”	21,681
#1 AND #2 AND #3 AND #4		170

**Table 6 ijerph-20-05085-t006:** Search strategy in CINAHL.

Acronyms	Search Strategy	Results
#1 Participants	“Palliative Care” OR “Hospice Care” OR (“Hospice and Palliative Care Nursing”) OR “Palliative Care Nursing” OR “Palliative Medicine” AND neoplasms OR tumor OR “Palliative Cancer Care” OR adult OR aged NOT child*	348,939
#2 Intervention	“Home Nursing” OR “Home Care Services” OR “Domiciliary Care” OR “Home Care” OR “Home Care Agencies” OR “Nursing Services” OR “Primary Health Care” OR “Primary Care” OR “Home Health Aide” OR “Public Health Practice” OR “Health Care, Primary” OR “Nonprofessional Home Care”	32,143
#3 Outcomes	“Quality of Life” OR “Health Related Quality Of Life” OR hrqol OR “Patient Acceptance of Health Care” OR “Health Care Acceptors” OR (“Organization and Administration”) OR “Disease Management” OR “Pain Management” OR “Financial Management” OR “Population Health Management” OR logistics OR supervision	112,587
#4 Study design to be included	“Non Randomized Controlled Trials as Topic” OR “Quasi-Experimental Study” OR “Non-Randomized Clinical Trial” OR “Randomized Controlled Trials as Topic” OR “Clinical Trials, Randomized” OR “Observational Study as Topic” OR “Observational Study” OR “Randomized Controlled Trials” OR “Non-Randomized Controlled Trials” OR “Observational Studies”	52,694
#1 AND #2 AND #3 AND #4		256

## Data Availability

No data were created.

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
