# Peer review of "The Organization of Home Palliative Cancer Care by Primary Health Care: A Systematic Review Protocol"

_ijerph, 2023, doi:10.3390/ijerph20065085_

Round 1
Reviewer 1 Report
Dear authors, I believe the manuscript should be improved in methodology, study design, introducing databases to search, discussing on availability of questionnaire, and separating inclusion and exclusion criteria.
Author Response
Response to Reviewer 1 Comments:
Point 1: I believe the manuscript should be improved in methodology, study design, introducing databases to search, discussing on availability of questionnaire, and separating inclusion and exclusion criteria.
Response 1: The aim and the research question were adjusted, and the method changes followed suit.

Reviewer 2 Report
Thanks for the opportunity to review this protocol paper. The topic is very important, and a systematic review on the field of palliative primary care would be very useful for researchers and clinicians tailoring care for these patients.
However, there are serious issues that preclude publication in this journal.
I find the general language style very hard to follow. There is the need to simplify expressions and concepts. I try to be clearer: I had to read the lines 75-79 four times before I could capture the concept. Normally, in a scientific paper, it is suggested to write short and simple phrases, in order to make it easier for a non-specialized reader understand the concepts. But I think that it is not the case for this manuscript.
The whole text needs intense proofreading in order to fix misspelling and typos.
Many sentences are not followed by appropriate referencing (e.g., 45-46; 68-69; 67-68).
There is inconsistency between what declared at the end of the introduction and what reported at the beginning of the methods. Apparently, there are more than one research questions to be answered.
There are macro-constructs that are not sufficiently addressed in the introduction: for example, what does exactly “primary health care coordination” mean? What about “care management”? In order to be investigated in a systematic review, these concepts should be clearly outlined and defined.
There is no sufficient mentioning of the outcomes reported in the strings, on the introduction.
The rationale for conducting the review is not outlined sufficiently. The authors claim that there are no evidence synthesis studies that focus on the organization of the primary palliative care and on the coordination of home palliative care. But this is not consistent with the aim of the review (impact of primary health care on the care management. Moreover, the fact that there are no studies addressing the topic is not sufficient to give a rationale for conducting any review.
It seems to me that the purpose of the study is not clear. Perhaps the care management of people with advanced cancer, coincides with the outcomes reported in the string? Very confusing.
The methodology of this systematic review is not in line with the aim. There is no mention of a meta-analysis to accomplish the evaluation of the impact of primary care on the patient outcomes. And also the subgroup analysis mentioned, is questionable. How do the authors achieve the identification of factors that hinder or contribute to the greater participation in primary health care in palliative cancer care? Moreover, this is another aim of the study it was never mentioned before.
Author Response
Response to Reviewer 2 Comments:
Point 1: Many sentences are not followed by appropriate referencing (e.g., 45-46; 68-69; 67-68).
Response 1: All these sentences were referenced.
Point 2: There is inconsistency between what declared at the end of the introduction and what reported at the beginning of the methods. Apparently, there are more than one research questions to be answered.
Response 2: Thanks for this valuable observation. We've simplified the aim and the research question so that for us they are aligned now.
Point 3: There are macro-constructs that are not sufficiently addressed in the introduction: for example, what does exactly “primary health care coordination” mean? What about “care management”? In order to be investigated in a systematic review, these concepts should be clearly outlined and defined.
Response 3: We totally agree with this note. We have included new references to support this macro-construct of the organization of home palliative cancer care by primary care. The term “care management” was excluded because of the change in aim.
Point 4: There is no sufficient mentioning of the outcomes reported in the strings, on the introduction.
Response 4: We include outcomes based on three new citations, which we believe contribute to the context of the introduction. We hope we have answered this question, but we are at your disposal if it is still not enough.
Point 5: The rationale for conducting the review is not outlined sufficiently. The authors claim that there are no evidence synthesis studies that focus on the organization of the primary palliative care and on the coordination of home palliative care. But this is not consistent with the aim of the review (impact of primary health care on the care management). Moreover, the fact that there are no studies addressing the topic is not sufficient to give a rationale for conducting any review.
Response 5: We understand the error of this statement, and we justify the review because there are different care models, with the application of different components for the integration of generalist and specialized palliative cancer care in primary care, and many countries need guidance to increase the supply of palliative care, in view of ever-increasing demand.
Point 6: It seems to me that the purpose of the study is not clear. Perhaps the care management of people with advanced cancer, coincides with the outcomes reported in the string? Very confusing.
Response 6: Based on the reflections of this reviewer, the aim has been changed.
Point 7: The methodology of this systematic review is not in line with the aim. There is no mention of a meta-analysis to accomplish the evaluation of the impact of primary care on the patient outcomes. And also the subgroup analysis mentioned, is questionable. How do the authors achieve the identification of factors that hinder or contribute to the greater participation in primary health care in palliative cancer care? Moreover, this is another aim of the study it was never mentioned before.
Response 7: We will seek to carry out a qualitative review, based on the different experiences of organizing care. We do not want to unite or combine results in a meta-analysis, but to highlight the best evidence in each model or experience, according to possible outcomes of home palliative cancer care organized by primary care. We revised item 2.2.2, and excluded the following intervention: Interventions for greater participation of primary health care in the management of patient care at home. We excluded subgroup analysis for this review.

Reviewer 3 Report
Thank you for allowing me to review your submission. Your paper can make significant contribution to the available evidence in your field of research.
However, to be engaging for the reader, the submission requires comprehensive editing. Please consider clear, concise, and succinct writing.

Author Response
Response to Reviewer 3 Comments:
Point 1: Abstract – revision of words choice and sentence structure.
Response 1: Thank you for the detailed review. We revise the sentences in the abstract, and we tried to use simpler language throughout the text. We remain available in case further changes are necessary.
Point 2: Introduction: Lines 33-35 – very long sentence, consider simple English language and breaking up. Line 45 - citation required. Line 32 – what disease? What practice? Line 56 – What stage? Lines 75-78 – Long sentence requires review and evidence support. Home care and primary care require clear definition.
Response 2: We revise the sentences and include references and definitions where requested.
Point 3: Line 104 – requires identification of clear end point.
Response 3: 2018-2022.
Point 4: Lines 257-259 – Citations requires.
Response 4: We include references.

Round 2
Reviewer 1 Report
Dear Editorial Team,
This manuscript is now acceptable.
All the best
Author Response
Thank you!

Reviewer 2 Report
Thanks for the points the authors have addressed. I have made quite a few editing on the manuscript to improve radability and flow. Moreover, I inserted also a few comments to recommend a revision of the introduction, which I find a bit in disorder. I suggested a strategy to follow-
Have a good work with the manuscript!

Author Response
Point 1: The introduction is a bit in disorder. I would structure the paragraphs with the same passages: 1) presenting palliative care and their benefits in general. 2) Underlying what is critical in general in the chronic population 3) report how the situation is in primary care; 5) The need for a review on the topic because the knowledge is heterogenous and fragmented.
Response 1: First of all, we would like to say that we greatly appreciate the grammar corrections. Thank you very much for this help. We edited the introduction as requested.
Point 2: Line 37: This is in conflict with what declared in the abstract where the authors claim that in developed countries, organization is good. Please align the ideas.
Response 2: In the abstract, we include the word ‘many’, and in the introduction ‘low- and middle-income countries’.
Point 3: Line 47: I would add the reason. I would not declare this. I would elaborate better on the advantage of investing more on cancer people because palliative care is more acceptable compared to other populations, and this is the main reason for limiting the review for these individuals and the organization caring for them.
Response 3: We revised this reason by articulating with the previous paragraph about the difficulties in performing palliative care, and also highlighting the epidemiology of cancer.
Point 4: Line 109: What do the authors mean by revising this definition? It is not in the aim of the review.
Response 4: We actually wanted to justify the beginning of the timeframe in 2018.
Point 5: Line 116: What about synthesizing the search strings and report all the strings for all the databases? Reporting only one database seems quite strange and unjustified.
Response 5: We have included tables with all the search strategies
Point 6: Line 130: Elderly are adults. Please edit.
Response 6: Edited.
Point 7: Line 131: Do they mean length of in-home care?
Response 7: Yes.
Point 8: Line 138: Making a paragraph with a single sentence is not suggested in scientific writing.
Response 8: Edited. But that was unavoidable in topic 2.2.5 – types of study.
Point 9: Line 141: Please reformat the bulleted list
Response 9: Formatted.
Point 11: Line 164: Outcomes do not comprise the attribute. So, for example: death rate, and not high death rate.
Response 11: Edited.
Once more, thank you very much!

Reviewer 3 Report
Thank you, I have reviewed your edits. Please ensure you thoroughly proofread prior to final submission, small word omission/grammatical errors remain.
Author Response
Point 1: Please ensure you thoroughly proofread prior to final submission, small word omission/grammatical errors remain.
Response 1: Thank you for your work. The text has been revised.
